# Usp10 Modulates the Hippo Pathway by Deubiquitinating and Stabilizing the Transcriptional Coactivator Yorkie

**DOI:** 10.3390/ijms20236013

**Published:** 2019-11-29

**Authors:** Yang Gao, Xiaoting Zhang, Lijuan Xiao, Chaojun Zhai, Tao Yi, Guiping Wang, Enlin Wang, Xiaohui Ji, Liangchang Hu, Guangshuang Shen, Shian Wu

**Affiliations:** The State Key Laboratory of Medicinal Chemical Biology, Tianjin Key Laboratory of Protein Sciences, College of Life Sciences, Nankai University, Tianjin 300071, China; gaoyangnankai@163.com (Y.G.); xiaolijuan1995@163.com (L.X.); zhaichaojun_0@163.com (C.Z.); 1410602@mail.nankai.edu.cn (T.Y.); 18202670817@163.com (G.W.); wangenlinnankai@163.com (E.W.); xiaohji@163.com (X.J.); huliangchang1987@126.com (L.H.)

**Keywords:** Hippo pathway, Yki, Usp10, deubiquitinating enzyme, *Drosophila*

## Abstract

The Hippo signaling pathway is an evolutionarily conserved regulator that plays important roles in organ size control, homeostasis, and tumorigenesis. As the key effector of the Hippo pathway, Yorkie (Yki) binds to transcription factor Scalloped (Sd) and promotes the expression of target genes, leading to cell proliferation and inhibition of apoptosis. Thus, it is of great significance to understand the regulatory mechanism for Yki protein turnover. Here, we provide evidence that the deubiquitinating enzyme ubiquitin-specific protease 10 (Usp10) binds Yki to counteract Yki ubiquitination and stabilize Yki protein in *Drosophila* S2 cells. The results in *Drosophila* wing discs indicate that silence of Usp10 decreases the transcription of target genes of the Hippo pathway by reducing Yki protein. In vivo functional analysis ulteriorly showed that Usp10 upregulates the Yki activity in *Drosophila* eyes. These findings uncover Usp10 as a novel Hippo pathway modulator and provide a new insight into the regulation of Yki protein stability and activity.

## 1. Introduction

As an evolutionarily conserved pathway from *Drosophila* to human, the Hippo signaling pathway plays critical roles in organ size control, homeostasis, and tumorigenesis [1,2,3]. The core components of the pathway form a kinases cascade, including Warts (Wts), Salvador (Sav), Hippo (Hpo), and Mob-as-tumor-suppressor (Mats), which are homologous to human large tumor suppressor 1 and 2 (LATS1/2), Salvador homolog 1 (SAV1), Mammalian Sterile 20-like kinases 1 and 2 (MST1/2), and MOB kinase activator 1 (MOB1). The Hpo-Sav kinase complex phosphorylates and activates the Wts-Mats kinase complex [4,5,6,7,8,9]. The primary target of this kinase cascade is the transcriptional coactivator Yorkie (Yki) (homologue to human protein YAP/TAZ) [4,6,7,10]. Yki transcriptionally promotes the expression of target genes by binding to the transcription factor Scalloped (Sd) (homologue to human protein TEAD1/2/3/4) in the nucleus [11,12]. The most well-known target genes of Yki-Sd are *cyclin E* (*cycE*) and *diap1*, which are involved in promoting cell proliferation and inhibiting apoptosis, respectively [9]. Inactivation of the kinases cascade, or ectopic expression of Yki, will lead to tissue overgrowth or tumorigenesis [13]. 

Although Yki is the primary target and key effector of the Hippo pathway, the regulatory mechanism of its protein stability is not completely clear. Recently, we found that the Yki protein may be eliminated by either lysosome or proteasome pathways in different cellular compartments, and the deubiquitinating enzyme (DUB) ubiquitin-specific protease 7 (Usp7) associates with and stabilizes Yki by promoting its deubiquitination [14]. Another DUB USP9X was reported to target YAP protein for deubiquitination and stabilization in mammals [15]. 

Approximately 80 functional DUBs were encoded in the human genome [16]. Ubiquitin-specific protease 10 (Usp10) is one of them, which plays important roles in many physiological and pathological processes, such as apoptosis [17,18,19], senescence [20], and tumorigenesis [21,22,23,24,25,26]. In *Drosophila*, Zhang et al. reported that Usp10 is a positive regulator of Notch signaling [27], and, most recently, Usp10 was implicated as a potential YAP-binding DUB among 68 human DUBs in HEK293T cells [28]. However, the functional relevance between Usp10 and YAP is totally unknown. Here, we used *Drosophila* S2 cells and animal models to investigate the possible relationship between Usp10 and Yki. Our results showed that Usp10 promotes Yki deubiquitination and stabilization through protein–protein interaction in S2 cells and silencing of Usp10 decreases the target genes’ expression by reducing Yki protein in *Drosophila* wing discs. Consistently, Usp10 also enhanced Yki activity in vivo in *Drosophila* eyes. Our studies revealed that Usp10 is a novel regulator in the Hippo signaling pathway and provided a new clue to further understand the regulatory mechanism of Yki protein stability and activity. 

## 2. Results

### 2.1. Ubiquitin-Specific Protease 10 (Usp10) Associates and Colocalizes with Yorkie (Yki) in the Cytoplasm

As mentioned above, the human Usp10 was reported as a potential YAP-binding protein [28]. However, the function of Usp10 in the Hippo signaling pathway still remains a mystery. In order to explore the relationship between *Drosophila* Usp10 and Yki, we generated a construct for expressing Myc-tagged Usp10-PA (Myc-Usp10-PA, the largest recognized isoform of Usp10 in flybase). Our immunoprecipitation (IP) assays showed that exogenously expressed Myc-Usp10 and HA-Yki were reciprocally co-immunoprecipitated (Figure 1A,B). In addition, our immunostaining assays further revealed that Usp10 colocalizes with and stabilizes Yki in the cytoplasm of S2 cells (Figure 1C–E”), suggesting that Usp10 might be a bona fide Yki-binding protein and can stabilize Yki by direct binding.

### 2.2. The Ubiquitin Carboxyl-Terminal Hydrolase (UCH) Domain of Usp10 Associates with Yki

*Drosophila* Usp10 mainly expresses three transcripts corresponding to two polypeptide isoforms: Usp10-PA/PC (1517aa) and Usp10-PB (797aa). Usp10-PB is identical to the C-terminal 797 amino acid residues of Usp10-PA (http://flybase.org/reports/FBgn0052479, Figure 2A). The Co-IP assays showed that exogenously expressed Myc-Usp10-PB and HA-Yki were also co-immunoprecipitated (Figure 2B). To further determine the specific binding region of Usp10 and Yki, we truncated Usp10-PB into C-terminal half (Usp10-PBC) containing the ubiquitin carboxyl-terminal hydrolase (UCH) domain and N-terminal half (Usp10-PBN) with no obvious domains (Figure 2A). From the Co-IP assays, we found that HA-Yki was precipitated with either Usp10-PA, Usp10-PB, or Usp10-PBC, but not with Usp10-PBN (Figure 2C), indicating that the UCH domain containing C-terminal of Usp10 is responding to associate with Yki specifically.

### 2.3. Usp10 Stabilizes Yki by Inhibiting the Proteasome-Mediated Degradation Pathway

As Usp10 functions as a ubiquitin-specific protease, we next examined whether Yki stability is regulated by Usp10. As expected, Usp10 increased the Yki protein level in a dosage-dependent manner (Figure 3A). We further confirmed that Usp10 promotes Yki protein accumulation by inhibiting Yki degradation through cycloheximide (CHX) chase assays (Figure 3B,B’). Conversely, *usp10* knockdown accelerated Yki turnover (Figure 3C,D). Furthermore, when *usp10* was depleted, the proteasomal inhibitor MG132 effectively reversed Yki destabilization, whereas the lysosomal inhibitor bafilomycin A1 (BA1) had no such effect (Figure 3E), suggesting that Usp10 stabilizes Yki by inhibiting protein decay through the proteasome-mediated degradation pathway.

### 2.4. Usp10 Promotes Yki Deubiquitination 

We next sought to explore whether Usp10-mediated stability regulation of Yki is dependent on the deubiquitination activity of Usp10. Our results showed that both Usp10-PA and Usp10-PB can markedly decrease the ubiquitination of Yki (Figure 4A). Intriguingly, we found that neither Usp10-PBN nor Usp10-PBC promoted Yki deubiquitination anymore (Figure 4B), suggesting that an intact Usp10-PB is required for its function on Yki deubiquitination. In contrast, Usp10 knockdown increased the ubiquitination of exogenously expressed HA-Yki (Figure 4C) or endogenous Yki (Figure 4D). Taken together, our results demonstrated that Usp10 regulates Yki stability by modulating Yki ubiquitination and proteasome-mediated degradation.

### 2.5. Usp10 Silence Decreases Yki Protein Level and Target Genes’ Expression In Vivo

To confirm that Usp10 regulates the protein stability of Yki in vivo, we knocked down *usp10* expression using *Drosophila* RNA-interference (RNAi) line (VDRC37859) to detect the protein level in *Drosophila* wing imaginal discs. The immunostaining results showed that *usp10*-RNAi driven by Hh-Gal4 (Hh > Usp10-RNAi) significantly reduced Yki protein level in the posterior region of wing discs, where Hh-Gal4 was expressed (Figure 5A,B), suggesting that Usp10 controls Yki turnover in vivo. As a transcriptional coactivator and the major effector in the Hippo signaling pathway, Yki associates with DNA-binding transcription factor Sd to transcriptionally promote the expression of target genes [10]. Consistent with Yki reduction upon *usp10* knockdown, the expression of DIAP1 and expended-lacZ (ex-lacZ) reporter, two well-known targets of Yki, are also dramatically decreased upon Hh > Usp10-RNAi in wing imaginal discs (Figure 5C–F), suggesting that Usp10 silence decreases Yki protein level and then its targets’ expression in vivo. To exclude the possibility of off-target, we generated a new Usp10-RNAi line (Usp10-shRNA), in which the target site is different from VDRC37859. Although this Usp10-shRNA is less effective, the Yki level and its targets’ (DIAP1 and ex-LacZ) expression are also significantly decreased in the Usp10-RNAi region, indicating that Usp10 is essential for maintaining Yki stability and its targets’ expression ( Appendix A).

### 2.6. Usp10 Regulates Yki Activity In Vivo

To further explore the functional relevance between Usp10 and Yki in vivo, we generated UAS-Usp10 transgenic flies, in which Usp10-PA could be induced to overexpress by tissue-specific Gal4 drivers. Our results showed that overexpression of Usp10 by eye-specific GMR-Gal4 driver (GMR > Usp10) resulted in shining, rough, and statistically bigger eyes compared to GMR-Gal4 control (Figure 6A,B,J, and Appendix A). In contrast, GMR > Usp10-RNAi (VDRC37859) led to reduced eye sizes and marginal patterning defects (Figure 6A,C,J, and Appendix A). Consistent with Usp10 regulating Yki stability and target gene expression in vitro and in vivo, GMR > Usp10 enhanced, while GMR > Usp10-RNAi suppressed, the enlarged eye phenotypes caused by GMR > Yki (Figure 6D–F,J, and Appendix A), indicating Usp10 as a positive regulator involved in modulating the stability and activity of Yki in vivo. In addition, we wondered whether Usp10 may also use a conserved mechanism to regulate the activity of mammalian YAP, the ortholog of *Drosophila* Yki. Human YAP-S127A has been reported as a constitutively active form, and transgenic flies expressing this YAP variant by the UAS/Gal4 system in *Drosophila* eyes (GMR > YAP-S127A) resulted in a dramatic increase in eye sizes [29] (Figure 6A,G, and Appendix A). Intriguingly, these enlarged eyes from GMR > YAP-S127A are enhanced further and suppressed by simultaneous expression of Usp10 and Usp10-RNAi, respectively (Figure 6G–J and Appendix A), suggesting that Usp10 modulating YAP activity may be conserved through protein stability regulation. 

In summary, our results provided strong evidence that Usp10 binds to Yki and regulates Yki activity and function through modulating the ubiquitination level and subsequent protein turnover. Our studies offer new insight into the molecular mechanism for the regulation of Yki protein stability and have potential implications for YAP-promoting tumorigenesis in humans.

## 3. Discussion

The Hippo signaling pathway regulates cell proliferation and apoptosis by modulating the activity of downstream effectors Yki (in *Drosophila*) or YAP/TAZ (in mammals), affecting the fate of cells, tissues, organs, and even organisms [10]. Previous studies have indicated that YAP can be degraded not only by the lysosomal pathway, but also by the ubiquitin-proteasome pathway [30,31,32]. As for the ubiquitin-proteasome pathway, YAP was reported to be ubiquitinated by E3 ligase complexes SCF^β-TRCP^ and SCF^FBXW7^ [31,33,34]. Recently, USP9X was reported to be a DUB for YAP deubiquitination, and promote tumor cell survival and chemoresistance [15]. However, the regulatory mechanism of Yki protein homeostasis in *Drosophila* is largely unknown. In this study, we reported that the deubiquitinase Usp10 positively regulates Yki stability and activity. Firstly, Usp10 associates and co-localizes with Yki in the cytoplasm of S2 cells, and thus, promotes Yki deubiquitination and stabilization via the proteasome-ubiquitination pathway. Secondly, Usp10 is required for maintenance of Yki stability, targets’ expression, and function output in *Drosophila* animals. Thirdly, Usp10 may use a conserved mechanism to modulate human YAP activity.

Yki protein turnover is mediated by both lysosome- and proteasome-dependent degradation pathways in *Drosophila* [14,35]. As first reported, the *Drosophila* deubiquitinase of Yki, Usp7, was also reported to positively regulate Yki stability and activity via deubiquitination. Thus, it will be interesting to further investigate the functional relevance between Usp10 and Usp7 on the regulation of Yki activity. Based on the fact that Usp7 was reported to localize mainly in the nucleus and stabilize nuclear Yki protein [14], while Usp10 mainly localizes with and stabilizes Yki in the cytoplasm (Figure 1C–E), we tend to believe that Usp7 and Usp10 may function in different subcellular compartments, and thus play collaborative or independent roles in controlling Yki stability and activity. 

Using *Drosophila* models, we have demonstrated that Usp10 maintains Yki activity by modulating its ubiquitination and turnover. Human Usp10 plays important roles in many physiological and pathological processes, including tumorigenesis [21,22,23,24,25,26]. However, which of and how these biological processes mediated by Usp10 rely on YAP and the Hippo pathway remains a mystery. Thus, it will be of importance to understand whether this regulatory mechanism is conserved in vertebrate or human beings. Our data simply show that the activity of human constitutive active YAP-S127A is also regulated by *Drosophila* Usp10 (Figure 6G–J and Appendix A), indicating that the conserved regulatory mechanism may be shared between human Usp10 and YAP. Indeed, human Usp10 was also reported as a YAP-binding partner [28]. Thus, further studies would be important to this scientific question and human healthcare.

## 4. Materials and Methods 

### 4.1. Plasmid Construction

To generate Myc-Usp10-PA and Flag-Ub expressing constructs, their corresponding coding sequences (CDS) were amplified from the cDNAs of *Drosophila* S2 cells and cloned into *pAc-V5/HisB* vector. Sequences encoding the N-terminal Myc epitope (MEQKLISEEDLNE) or Flag epitope (MDYKDDDDK) were added by PCR in place of the first Met codon of the respective CDS. Constructs of Myc-Usp10-PB, Myc-Usp10-PBN, and Myc-Usp10-PBC in *pAc-V5/HisB* vector were generated using Myc-Usp10-PA as the template. No tagged Usp10-PA was cloned into *pUAST-attB* vector by PCR for Usp10-PA expression in vivo. *pAc-HA-Yki* has been described previously [10]. All generating constructs in this study were confirmed by DNA sequencing.

### 4.2. Cell Culture, Transfection, and Drugs Treatment

*Drosophila* S2 cells were cultured in serum-free insect cell culture medium (Gibco, WA, Massachusetts, USA) with 10% fetal bovine serum (FBS, Hyclone, Logan, UT, USA) at 27 °C without CO_2_. Transient transfection was performed using X-treme GENE HP DNA transfection reagent (Roche, Basel, Switzerland), according to the manufacturer’s instructions. Cells were harvested after transfection for 36 h. For protein stability assays, a final concentration 20 µM of proteasome inhibitor MG132 (Selleckchem S2619, Houston, TX, USA) or 20 nM of lysosome inhibitor bafilomycin A1 (BA1; Abcam ab120497, Cambridge, UK) was added to culture medium for 4 h before S2 cells collection, and 75 µg/mL protein synthesis inhibitor cycloheximide (CHX; Sigma B01075503, Kanagawa, Japan) was used to treat S2 cells at indicated intervals before harvesting.

### 4.3. RNA Interference

To knockdown Usp10 expression in S2 cells, two dsRNAs targeting different regions of Usp10 were synthesized with the T7 RiboMAX™ express RNAi System (Promega, Madison, WI, USA) according to the manufacturer’s instructions. The templates for dsRNA synthesis were PCR-amplified from genome DNA of S2 cells by the following primer pairs: 

dsRNA #1F:

5′-GAATTAATACGACTCACTATAGGGAGACCAGTTCATCAGCCAATCC-3′,

dsRNA #1R:

5′-GAATTAATACGACTCACTATAGGGAGACTTGTGGCGCGTCAGG-3′;

dsRNA #2F:

5′-GAATTAATACGACTCACTATAGGGAGAAGAAGACCTCGCAGAAGCAG-3′,

dsRNA 2R:

5′-GAATTAATACGACTCACTATAGGGAGATCACACGCACTGACACAAAA-3′.

S2 cells were seeded into 6-well tissue culture plates incubated with 15 μg dsRNA in the medium and cultured for 24 h, then 1 mL fresh medium containing 10 μg dsRNA was added, and the mixture was cultured for another 24 h before harvesting. dsRNA targeting the full coding sequence of GFP was used as a negative control.

### 4.4. Immunoprecipitation, Immunoblotting, and Immunofluorescence

S2 cells were transfected with the indicated plasmids and cultured for 36 h, then cells were collected and lysed in NP-40 lysis buffer plus protease inhibitors (Roche) for 30 min on a rotor at 4 °C. Immunoprecipitations (IP) and immunoblotting (IB) were performed as in previous reports [11]. Immunoreactive bands were visualized with chemiluminescence kits (Vazyme, Nanjing, China) on the Tanon-5500 Chemiluminescent Imaging System (Tanon Science and Technology, Shanghai, China). Antibodies used in IP and IB assays: mouse anti-Myc (Utibody, Tianjin, China, 1:200 for IP and 1:5000 for IB), rabbit anti-Myc (Proteintech, 1:500 for IB), mouse anti-HA (Abmart, Shanghai, China, 1:200 for IP and 1:5000 for IB), rabbit anti-HA (CST, Hopkinton, MA, USA, 1:5000 for IB), rabbit anti-Flag (Abmart, 1:500 for IB), mouse anti-Tubulin (Utibody, 1:10,000 for IB), rabbit anti-Yki (gift from Dr. Duojia Pan, 1:200 for IP and 1:5000 for IB), and mouse anti-Ub (Santa cruz, CA, USA, 1:500 for IB). 

Immunofluorescence (IF): Cells were transfected with the indicated plasmids after being seeded on coverslips. Thirty-six hours later, cells were washed by PBS and fixed in 4% paraformaldehyde for 20 min and permeabilized with 0.5% Triton X-100 for 30 min at room temperature. Cells were then blocked for 30 min in 3% bovine serum albumin (BSA) and incubated with primary antibody for 2 h, then fluorophore-conjugated secondary antibody for another 1 h at room temperature. DAPI was used to mark nuclear for 6 min before the slides were mounted. Cell images were captured with confocal microscope (Leica, Wetzlar, Germany). Antibodies used for IF: mouse anti-Myc (Utibody, 1:1000), rabbit anti-HA (CST, 1:1000), Alexa Fluor 488 goat anti-mouse IgG (Invitrogen, Carlsbad, CA, USA, 1:1000), and Alexa Fluor Cy3 goat anti-Rabbit IgG (Invitrogen, 1:1000).

### 4.5. Immunostaining of Wing Imaginal Discs

Third-instar larvae were dissected in PBST (PBS plus 0.3% Triton X-100) and fixed in 4% paraformaldehyde at room temperature for 60 min, then washed three times for 20 min each time with PBST. Larvae were then blocked for 1 h in PBST with 5% FBS and incubated overnight with primary antibodies in PBST with 5% FBS at 4 °C, then washed three times for 20 min each time and incubated with corresponding fluorophore-conjugated secondary antibody for 2 h at room temperature. After washing three times in PBST, discs were dissected and mounted. Disc’s images were captured with a confocal microscope (Leica). Antibodies used were as follows: mouse anti-DIAP1 (gift from Dr. DJ Pan, 1:200), mouse anti-βGal (Promega, 1:200), rabbit anti-Yki (gift from Dr. DJ Pan, 1:500), Alexa Fluor Cy3 goat anti-mouse IgG (Invitrogen, 1:1000), and Alexa Fluor Cy3 goat anti-Rabbit IgG (Invitrogen, 1:1000).

### 4.6. Drosophila Stocks

All fly hybridization experiments were performed at 25 °C. The stock of Usp10-RNAi (37859) belongs to Vienna *Drosophila* Resource Center (VDRC). The UAS-Usp10 transgenic flies were generated by injection of *pUAST-attB-Usp10* constructs into *Drosophila* embryos. The Usp10*-*shRNA transgenic flies were generated by injection of *UASp-Usp10-*shRNA constructs into *Drosophila* embryos. The following sequences were used:

Usp10*-*RNAi-1F: 5′-CTAGCAGTGATTTGCAACAATCGTAATAATAGTTATATTCAAGCATATTATTACGATTGTTGCAAATCGCG-3′

Usp10-RNAi-1R: 5′-AATTCGCGATTTGCAACAATCGTAATAATATGCTTGAATATAACTATTATTACGATTGTTGCAAATCACTG-3′

The following flies have been described: *UAS-yki* [10] and *UAS-yap*-S127A [29]. Hh-gal4, ex-lacZ, and GMR-gal4 were gifts from Dr. DJ Pan’s lab. The other stocks are stored in our stock collections. The images of eyes were captured with Stereo Fluorescence Microscope (Leica).

### 4.7. Statistical Analysis

The density of the IB band was measured by Image J software. Statistical analysis was performed with GraphPad Prism software; the error bars indicate standard deviation. Significance was determined by the two-tailed, unpaired Student’s *t*-test, and *p* < 0.05 was considered statistically significant.

## Figures and Tables

**Figure 1 ijms-20-06013-f001:**
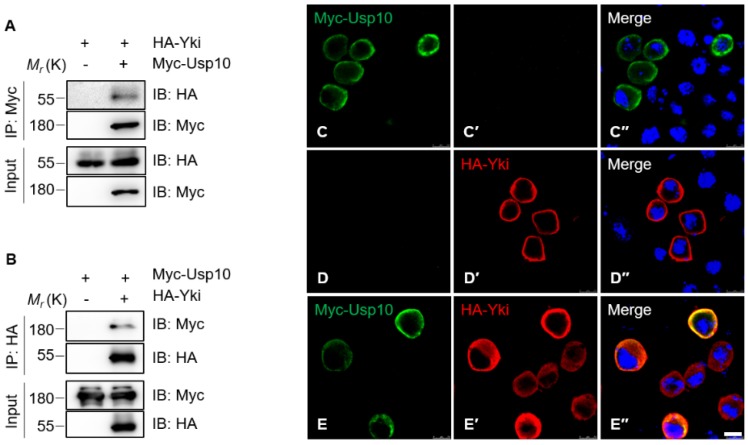
Ubiquitin-specific protease 10 (Usp10) associates and colocalizes with Yorkie (Yki) in the cytoplasm of S2 cells. Co-immunoprecipitation of exogenously expressed HA-Yki with (**A**) Myc–Usp10-PA and (**B**) vice versa. S2 cells were transfected with plasmids for expressing (**C**–**C**′′) Myc-Usp10-PA or (**D**–**D**′′) HA-Yki alone, or (**E**–**E**′′) Myc-Usp10-PA together with HA-Yki, and subjected to immunostaining with the indicated anti-tag antibodies. Images were collected by confocal microscopy. Scale bars: 7.5 μm.

**Figure 2 ijms-20-06013-f002:**
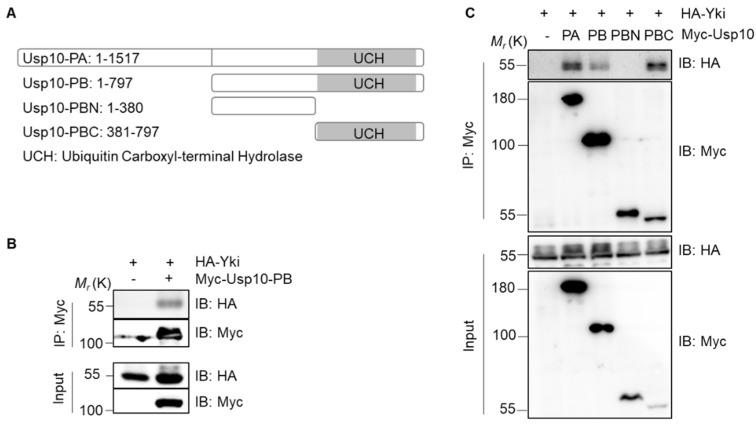
The ubiquitin carboxyl-terminal hydrolase (UCH) domain of Usp10 associates with Yki in S2 cells. (**A**) The scheme of proteins Usp10-PA, Usp10-PB and their truncations. (**B**) Co-immunoprecipitation (Co-IP) of exogenously expressed HA-Yki with Myc–Usp10-PB. (**C**) Co-immunoprecipitation of exogenously expressed HA-Yki with Myc–Usp10 PA/PB/PBN/PBC.

**Figure 3 ijms-20-06013-f003:**
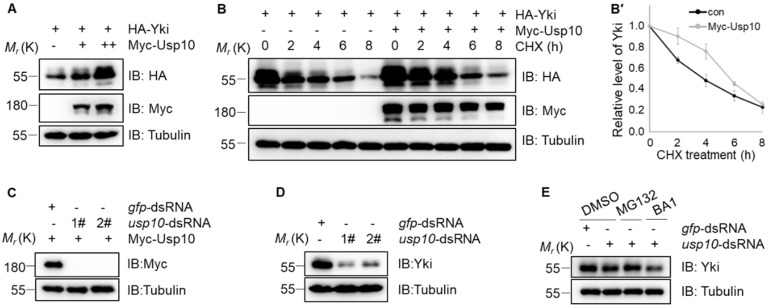
Usp10 regulates Yki stabilization through the proteasome-mediated degradation pathway in S2 cells. (**A**) Cells expressing HA-Yki and gradient Myc-Usp10-PA were collected for immunoblotting with the indicated antibodies. (**B**) Cells expressing HA-Yki with or without Myc-Usp10-PA were treated with 75 μg/mL cycloheximide (CHX) for indicated intervals and collected for immunoblotting. (**B**′) Quantification analysis of Yki protein level for B (normalized to Tubulin). Data here are the mean of *n* = 3 independent experiments, and error bars indicate standard deviation (SD). (**C**) Cells expressing Myc-Usp10-PA were used for the knocking down efficiency test of double-strand RNAs (dsRNAs) targeting to *usp10* or *gfp*. (**D**) Cells knocking down of *gfp* or *usp10* with dsRNAs were collected for immunoblotting with indicated antibodies. (**E**) Cells incubated with dsRNA of *gfp* or *usp10* were treated with DMSO, MG132, or bafilomycin A1 (BA1) for 4 h before collection for immunoblotting.

**Figure 4 ijms-20-06013-f004:**
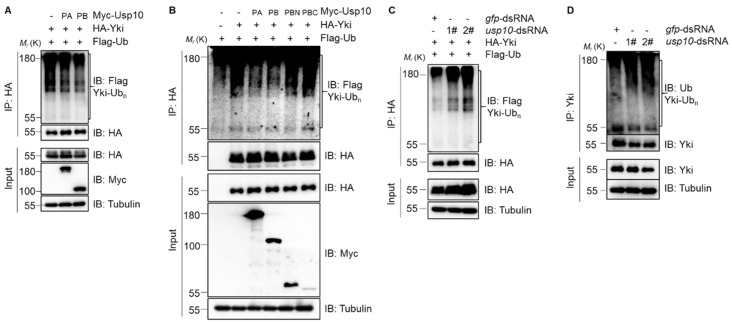
Usp10 regulates the ubiquitination level of Yki in S2 cells. Cells expressing indicated proteins were treated with 20 μM MG132 for 4 h before harvesting, and the immunoprecipitates or lysates were submitted for immunoblotting with the indicated antibodies. (**A**,**B**) Cells expressing Myc-Usp10 reduce the ubiquitination level of HA-Yki. Usp10-PA and Usp10-PB effectively deubiquitinate HA-Yki, while the truncations Usp10-PBN and Usp10-PBC do not. (**C**,**D**) Usp10 knocking down promotes the ubiquitination of (**C**) HA-Yki and (**D**) endogenous Yki.

**Figure 5 ijms-20-06013-f005:**
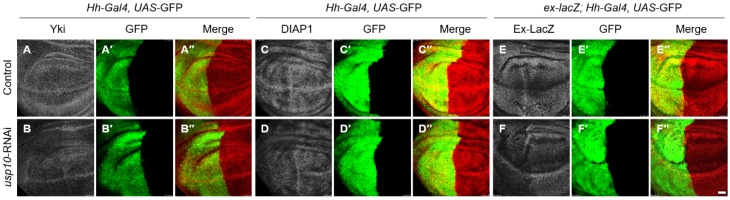
Usp10-RNA-interference (RNAi) reduces Yki protein level and target genes’ expression. Shown are wing imaginal discs from third instar larvae with posterior to left and notum up. The scale bars are 50 μm. (**A**–**A**′′) Representing wing imaginal discs of control (Hh-Gal4, UAS-GFP/+) were dissected and immunostained to show the expression patterns of Yki (white in **A** and red **A′′**) and GFP (green in **A′**–**A′′**). GFP indicates the expression pattern of Hh-Gal4. (**B**–**B**′′) Representing wing imaginal discs of Usp10 knockdown (Hh-Gal4, UAS-GFP/UAS-Usp10-RNAi) were dissected and immunostained to show the expression patterns of Yki (white in **B** and red in **B′′**) and GFP (green in **B′**–**B′′**). (**C**–**C**′′ and **D**–**D**′′) Representing wing imaginal discs of control (Hh-Gal4, UAS-GFP/+) (**C**–**C′′**) and Usp10 knockdown (Hh-Gal4, UAS-GFP/UAS-Usp10-RNAi) (**D**–**D′′**) were dissected and immunostained to show the expression patterns of DIAP1 (white in **C** and **D**, red in **C′′** and **D′′**) and GFP (green in **C′**–**C′′** and **D′**–**D′′**). (**E**–**E**′′ and **F**–**F**′′) Representing wing imaginal discs of control (ex-LacZ/+; Hh-Gal4, UAS-GFP/+) (**E**–**E′′**) and Usp10 knockdown (ex-LacZ/+; Hh-Gal4, UAS-GFP/UAS-Usp10-RNAi) (**F**–**F′′**) were dissected and immunostained to shown the expression patterns of ex-LacZ (white in **E** and **F**, red in **E′′** and **F′′**) and GFP (green in **E′**–**E′′** and **F′**–**F′′**).

**Figure 6 ijms-20-06013-f006:**
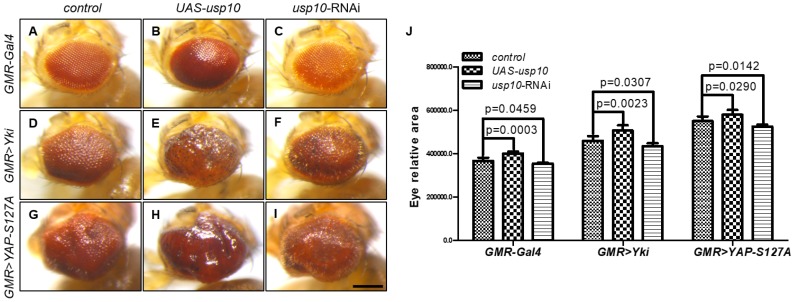
Usp10 regulates Yki activity in vivo. (**A**–**I**) Shown are images of the side view of adult male fly eyes; the scale bars are 200 μm. The genotypes are listed: GMR-Gal4/+ (**A**), GMR-Gal4/+; UAS-Usp10/+ (**B**), GMR-Gal4/+; UAS-Usp10-RNAi/+ (**C**), GMR-Gal4, UAS-Yki/+ (**D**), GMR-Gal4, UAS-Yki/+; UAS-Usp10/+ (**E**), GMR-Gal4, UAS-Yki/+; UAS-Usp10-RNAi/+ (**F**), GMR-Gal4, UAS-YAP-S127A/+ (**G**), GMR-Gal4, UAS-YAP-S127A/+; UAS-Usp10/+ (**H**), GMR-Gal4, UAS-YAP-S127A/+; UAS-Usp10-RNAi/+ (**I**). Quantitative analysis of relative eye area from **A**–**I** is shown (*n* = 7) (**J**), and error bars indicate SD. Significance is determined by two-tailed, unpaired Student’s *t*-test.

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
