# Peer review of "Usp10 Modulates the Hippo Pathway by Deubiquitinating and Stabilizing the Transcriptional Coactivator Yorkie"

_ijms, 2019, doi:10.3390/ijms20236013_

Round 1

Reviewer 1 Report

The authors have addresses all concerned raised previously. The manuscript is has improved in quality and conclusions.

In the opinion of this reviewer, the manuscript should be accepted for publication.

Reviewer 2 Report

This is the second time I read the manuscript entitled: Ups10 modulates the Hippo pathway by deubiquitinating and stabilizing the transcriptional coactivator Yorkie by Gao et al.

The authors have made a big effort to address all my comments and the manuscript reach now the standard needed for publication.

Author Response

This manuscript is a resubmission of an earlier submission. The following is a list of the peer review reports and author responses from that submission.

Round 1

Reviewer 1 Report

In their paper Gao et al. address the question if Usp10, a deubiquitinating enzyme, stabilizes the Hippo pasthway component Yorkie. Using Co-immunoprecipitation they demonstrate that Usp10 is associated with Yorkie and also colocalized with Yorkie in the cytoplasm. They map the UCH domain as the interacting part of Usp10. Furthermore they showed that the stabilization of Yorkie is the result of an inhibition of the proteasome-mediated degradation pathway and that Usp10 is promoting Yorkie deubiquitination. Using ectopic expression and RNAi of Usp10 they demonstrate that the the Yorkie target genes diap1 and cycE could be upregulated by overexpression or downregulated by depletion of Usp10. Finally they analyse effects of Usp10 overexpression or downregulation in vivo in Drosophila using the UAS-Gal4 system. Here they show some effects in the eyes using GMR-Gal4 and in the wing using Nab-Gal4.

They data are well presented and justify the conclusions drawn except for the in vivo analysis in Drosophila (Fig.6). First of all it is not mentioned where the YAPS127A fly strain came from or how he was made and which effect the serine to alanine mutation has in gewneral. The other strain used (strain 15878) is a P-element insertion in the usp10 gene region (EY02707) harbouring some UAS sequences, but to my opinion not a bona fide UAS-usp10 strain, is it shown that usp10 is indeed overexpressed using this strain? The insertion of the P-element could also have a general effect on the expression of usp10 or lead to lethality, is this known? Also for the UAS-usp10 RNAi strain 36897 I could not find anything at the Bloomington website, where does this strain come from? Before these points are not clarified, I am not sure if the results presented in Fig.6 are correct.

In Fig1. the colocalization of Usp10 and Yorkie is shown. Why are so few cells expressing both proteins. I would also expect some cells expressing one or the other component and some cells both proteins. Could you comment on that.

In the sentence  from line 142-144 are some mistakes, in line 148 it should be „were significantly enhanced“.

Reviewer 2 Report

In this manuscript, Gao et. al. present data to suggest a new role for Usp10 in modulating Hippo pathway.  Overall the authors present data that supports a as yet unrecognized role for Usp10 in deubiqutinating and stabilizing Yorkie- a key effector of the Hippo growth control pathway.

The authors first show that Usp10 can associate with Yki and they colocalize in Drosophila S2 cells. They map the association to the UCH domain of Usp10 and show that Usp 10 inhibits proteasome-mediated degradation of Yki thereby stabilizing it.The mechanism of this interaction revealed that Usp10 promotes deubiquitination of Yki. Intriguingly Usp10 also affects Yki activity as overexpression of Usp10 caused upregulation of Yki transcriptional targets, and downregulation of Usp10 reduced the transcriptional output of Yki in RT-PCR and genetic interaction experiments. Based on these data, the authors propose that Usp10 functions as new regulator of Yki activity by physically associating with Yki and stabilizing Yki by controlling its deubiquitination.

Overall the experiments are well planned, the data is well presented and the conclusions follow from the data. However,  if the authors address the comments/critiques below it will help improve the manuscript and benefit the readership of International Journal of Mol. Sci. 

Major comments: 

It is interesting that Usp10 belongs to a family of proteins of which Usp7 and 9 are already reported to associate with Yki/YAP  and stabilize it by promoting deubiquitination- a mechanism that is shown for Usp10 by the authors. This raises an important questions: 

a. Can Usp7 and Usp10 act redundantly or do they play independent role in controlling Yki activity?

b. What is the genetic and molecular interaction between Usp7 and Usp10? If these proteins all act in the same complex?

The authors show by RT PCR that diap1 and CycE  expression is regulated by Usp10 presumably by its effects on Yki stability. However, neither diap1 nor CycE are exclusive transcriptional targets of Yki. The authors should use in vivo experiments like effect of Usp10 overexpression on ex-lacZ expression.

Minor comments:
Please check for English (grammatical inconsistencies) throughout the manuscript.

Reviewer 3 Report

At the first glance, the paper looks attractive and of wide interest as it addresses the question of the regulation of Yorkie, a key component of the now famous Hippo signaling pathway. In the line of their previous work on USP7 (Sun et al., 2019), the authors identified the deubiquitinating enzyme Usp10 as a potential interactor/regulator of Yorkie. The story is based on classical set of experiments from biochemistry to genetic interaction.

As I said earlier if the topic is of interest, the way it is address is not following the standards required for publication.

First, there are lots of typo, mistakes & mislabeling in figures 1A, 1B, 2B. This accumulation of mistakes greatly reduces confidence in the results presented.

 Second, for the genetic interaction between Usp10 and Yki (Figure 6) the authors used an Usp10-RNAi line from the Bloomington stock center. The reference of this line did not exist. In addition, two strains are available at the VDRC stocks center (V37858 and V37859) and have been already used to deplete the function of Usp10 (Zhang et al., G3, 2012). The expression of these 2 VDRC’s RNAi lines in the wing induces a strong phenotype. Even if Gal4 drivers are different in both cases, the authors should have seen a defect since nub expression cover C96-Gal4 expression. The VDRC lines can also generate a defect in the eye when expressed with GMR-Gal4 (Zhang et al., G3, 2012). Once again this is in contradiction with results presented here (with the same driver this time).

Third, the authors claim to have used an UAS-Usp10. In reality they used an EP lines, an insertion in the locus of Usp10 that should in theory allow the misexpression of the gene. It is important to control this point and/or generated proper lines from the cDNA of Usp10 as they did for the cell experiments.

Minors points:

Fig3A: Usp10 is not easy to see in line 2.

Fig3B: It would be nice to perform the CHX treatment up to 8h.

Fig3D: The decrease of Yki is not very convincing.

It would be nice to check whether the level of Yki is affected in vivo by the lack of USP10.

Fig5: It would be nice to check whether the level of diap1 and/or cycE are affected in vivo by the lack of USP10.

Fig6: The results obtained in the wing analysis are not very convincing once again. The authors should perform this experience in a density control condition.

The authors must site the paper of Zhang et al. (Zhang et al., A Targeted In Vivo RNAi Screen Reveals Deubiquitinases as New Regulators of Notch Signaling, G3, 2012)